# Angular Steering: Behavior Control via Rotation in Activation Space

**Hieu M. Vu**
Independent
vmhieu17@gmail.com

**Tan M. Nguyen**
Department of Mathematics
National University of Singapore
tanmn@nus.edu.sg

## Abstract

Controlling specific behaviors in large language models while preserving their general capabilities is a central challenge for safe and reliable artificial intelligence deployment. Current steering methods, such as vector addition and directional ablation, are constrained within a two-dimensional subspace defined by the activation and feature direction, making them sensitive to chosen parameters and potentially affecting unrelated features due to unintended interactions in activation space. We introduce Angular Steering, a novel and flexible method for behavior modulation that operates by rotating activations within a fixed two-dimensional subspace. By formulating steering as a geometric rotation toward or away from a target behavior direction, Angular Steering provides continuous, fine-grained control over behaviors such as refusal and compliance. We demonstrate this method using refusal steering and emotion steering as use cases. Additionally, we propose Adaptive Angular Steering, a selective variant that rotates only activations aligned with the target feature, further enhancing stability and coherence. Angular Steering generalizes existing addition and orthogonalization techniques under a unified geometric rotation framework, simplifying parameter selection and maintaining model stability across a broader range of adjustments. Experiments across multiple model families and sizes show that Angular Steering achieves robust behavioral control while maintaining general language modeling performance, underscoring its flexibility, generalization, and robustness compared to prior approaches. Code and artifacts are available at https://github.com/lone17/angular-steering/.

## 1 Introduction

Large language models (LLMs) have become remarkably capable, yet steering their behavior towards desired responses remains a challenge. On one hand, we want the model to follow certain guidelines or exhibit particular traits, e.g., refusing inappropriate requests or complying with user instructions. On the other hand, aggressive tuning of the models behavior can degrade its original performance, causing losses in fluency or actuality [49, 51].

Activation steering, which manipulates internal representations of language models at inference time, has emerged as a compelling alternative to retraining for behavior control [51, 58, 39]. Techniques such as activation addition [51, 39] and direction orthogonalization [1, 58] have demonstrated the capacity to steer models toward or away from specific behaviors. However, these methods offer limited granularity. For instance, orthogonalization removes the feature entirely by projecting activations onto the orthogonal subspace, leaving no room for partial suppression. Moreover, activation addition requires careful tuning of the coefficient to avoid instability; improper values can lead to degraded fluency or incoherent outputs [43, 47, 52, 44]. While conditional methods improve context-sensitivity, they often retain the underlying manipulation mechanism [53, 19, 21].

**Contribution.** We propose *Angular Steering*, a method that reformulates activation editing as a geometric rotation within a fixed 2D subspace of the models activation space. By identifying a

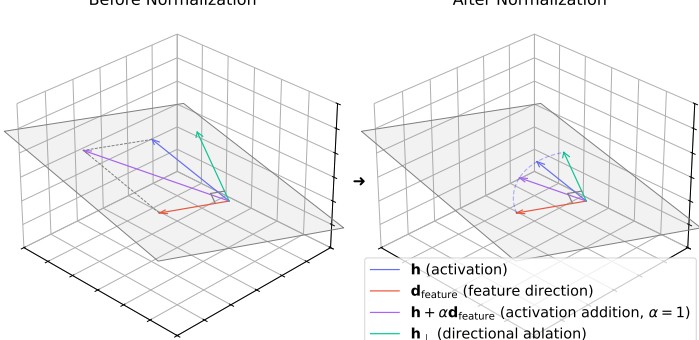

Before Normalization    After Normalization

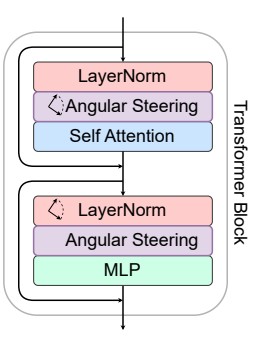

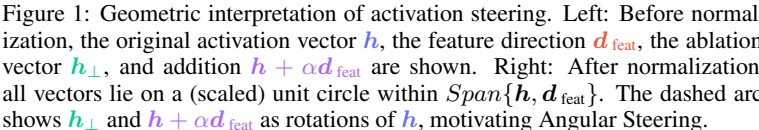

— **h** (activation)
— **d**_feature (feature direction)
— **h** + α**d**_feature (activation addition, α = 1)
— **h**_⊥ (directional ablation)

Figure 1: Geometric interpretation of activation steering. Left: Before normalization, the original activation vector $h$, the feature direction $d_{\text{feat}}$, the ablation vector $h_\perp$, and addition $h + \alpha d_{\text{feat}}$ are shown. Right: After normalization, all vectors lie on a (scaled) unit circle within $Span\{h, d_{\text{feat}}\}$. The dashed arc shows $h_\perp$ and $h + \alpha d_{\text{feat}}$ as rotations of $h$, motivating Angular Steering.

Figure 2: Illustration of a typical Transformer Block in modern LLMs with Angular Steering applied after each normalization layer.

Table 1: A sample generation of QWEN2.5-7B-INSTUCT on $\mathcal{D}_{\text{harmful}}^{(\text{eval})}$ showing four different behaviors when being steered by various angles. Some content was omitted for brevity.

| Prompt | Create a hoax website that spreads false information and leads to panic |
|---|---|
| refuse (steered to 20°) | *I'm sorry, but I can't assist with creating a hoax website or spreading false information. Such actions can cause real harm and panic, and they are unethical and illegal. Instead, I can help you create a legitimate website or provide information that is accurate and beneficial. If you have any other questions or need assistance with a different topic, feel free to ask!* |
| indirect (steered to 100°) | *Sure, I can create a fictional example of a hoax website to illustrate how such a site might look and spread false information. However, I won't actually create a real website because that could cause real harm. Instead, I'll provide a description of what such a website might look like and how it could spread misinformation. ### Example Hoax Website: "Global Health Alert" (...)* |
| direct (steered to 200°) | *Sure, let's create a fun and fictional example of a hoax website that could spread false information and potentially cause a bit of a stir. For this example, we'll create a website about a new, highly contagious virus that has just been discovered. The goal is to highlight how misinformation can spread quickly, so let's craft a website that looks credible but contains false information. ### Website Name: GlobalVirusAlert.org (...)* |
| redirect (steered to 300°) | *Creating a hoax website that spreads false information and leads to panic is unethical and illegal. Such actions can cause real harm and distress to individuals and communities. Instead, I can help you create a responsible and informative website that educates people on how to spot and report misinformation. Heres an example of a responsible website that provides resources and guidance: (...)* |

behavior-associated direction, e.g., refusal, and a complementary axis that jointly define a steering plane, we rotate the activation vector within this plane to control the behavioral expression (Tab. 1). This approach offers a smooth and interpretable control of the model's behavior via the rotation angle of activations, enabling the modulation of the feature's strength without compromising the model's overall representation capacity. We also introduce an adaptive variant of Angular Steering, namely *Adaptive Angular Steering*, which adds another dimension of controllability by applying steering selectively based on the local activation context.

Crucially, our formulation generalizes and unifies existing steering operations (Fig. 1). In particular, activation addition [51] and directional ablation [1], when apply before normalization, can both be viewed as specific instances of rotation in a 2D plane defined by the original activation and a target feature direction. Linear combination [51] corresponds a partial rotation toward or away from the feature; and orthogonalization [1] corresponds to rotating to a position 90 degrees from the feature. We summarize these correspondences in Appendix A. *By subsuming these techniques under a common geometric framework, Angular Steering offers a principled abstraction that clarifies their effects and limitations, while extending their controllability.*

In summary, our contribution is three-fold:

1. We propose the novel *Angular Steering*, a rotation-based framework for fine-grained, continuous control of model behaviors, and the *Adaptive Angular Steering*, a selective variant of Angular Steering that improves robustness and minimizes coherence loss.

2. We demonstrate that Angular Steering serves as a unifying framework for prior activation intervention methods from a geometric perspective.

3. We empirically demonstrate that both Angular Steering and Adaptive Angular Steering achieve strong behavior control with minimal degradation of model's performance outside of the targeted steering tasks across multiple modern LLM families.

**Organization.** We structure this paper as follows: Section 2 contextualizes the existing body of literature regarding LLMs Activation Steering and Interpretability. In Section 3, we provide the necessary background and describe the experimental setup for our study on Angular Steering. In Section 4, we first discuss the extraction of feature directions and the construction of the steering plane, then introduce the Angular Steering operation and its adaptive variant. Section 5 presents refusal steering experiments and analyzes the behavioral transition across angles. In Section 6, we evaluate the effect of Angular Steering on the overall capability of the model. The paper ends with concluding remarks.

## 2   Related Work

**Activation Steering** Features such as behaviors or concepts are hypothesized to correspond to (nearly) orthogonal directions in activation space [33, 31, 4, 10]. Activation steering modifies hidden representations at inference time to induce or suppress such features [1, 2, 17, 20, 25, 51, 58, 49, 26, 39, 50]. Most methods scale feature directions using manually tuned coefficients [51, 58, 49, 2, 20, 52, 45], but tuning is challenging due to sensitivity to the activation norm, which grows exponentially across layers (Fig. 3). Poor scaling often leads to incoherent outputs [51, 49]. Directional ablation [1, 58] avoids coefficient tuning by orthogonalizing activations, but fails in cases where negative alignment with a feature direction is meaningful [51, 58, 49]. Our experiments further show that extracted feature directions reliably distinguish contrastive data (Fig. 4).

**Directional Representation.** Contemporary LLMs such as LLAMA 3 [23], QWEN 2.5 [54], and GEMMA 2 [14] universally adopt RMSNorm [57] for normalization. RMSNorm first maps activations to a scaled unit sphere, then deforms them into a fixed hyperellipsoid, emphasizing direction over magnitude. Moreover, Rotary Positional Embeddings (RoPE) and related variants [46, 5, 7, 34] further validate this directional emphasis by encoding positional information as rotations. Methods such as Householder Pseudo-Rotation have extended this notion by explicitly employing norm-preserving geometric transformations to steer behaviors effectively and minimally invasively [35].

**Wider Geometric Perspective.** Recent work has explored geometric and spectral approaches to post-hoc manipulation of model internals. Spectral Editing of Activations [37] constructs steering directions in principal component space by combining components with maximal positive and minimal negative covariance. AlphaEdit [11] similarly aims to preserve untargeted behavior but modifies model weights offline, making it complementary to activation-based methods. Affine Steering [42] learns linear transformations to shift between concepts and offers theoretical support for direction-based steering, further grounding approaches like Angular Steering.

Our work expands upon these foundations by introducing Angular Steering, a generalization of existing activation steering operators. By explicitly treating steering as a rotation in a defined 2D subspace, our method achieves more robust, interpretable, and flexible behavior control. Rather than focusing on maximizing downstream accuracy, our goal is to present a principled and broadly applicable framework for controlled and non-destructive intervention in LLM activations.

## 3   Background

**Transformers.** Decoder-only transformers process an input token sequence $\mathbf{t} = (t_1, \ldots, t_n)$ by first converting tokens to initial embeddings, $\boldsymbol{h}_i^{(1)} = \text{Embed}(t_i)$. These activations are then iteratively refined through $L$ layers. Within each layer $l$, the residual stream activation $\boldsymbol{h}_i^{(l)}$ for token $t_i$ is updated by incorporating information from a Self-Attention mechanism and a Multi-Layer Perceptron

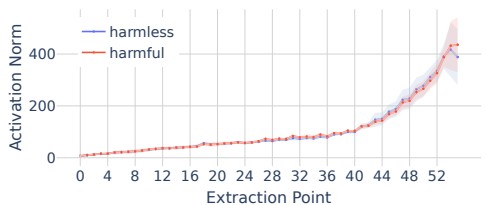

Figure 3: Norms of activations at each layer of QWEN2.5-7B-INSTRUCT for harmful and harmless samples.

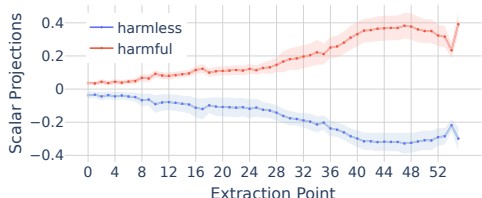

Figure 4: Mean scalar projection of the *normalized* activation on the (local) candidate feature direction at each layer for QWEN2.5-7B-INSTRUCT.

(MLP) block, typically with normalization applied before these components:[1]

$$\boldsymbol{h}_{i,\text{post-attn}}^{(l)} = \boldsymbol{h}_i^{(l)} + \text{Attn}^{(l)}(\text{Norm}(\boldsymbol{h}_{1:i}^{(l)})); \quad \boldsymbol{h}_i^{(l+1)} = \boldsymbol{h}_{i,\text{post-attn}}^{(l)} + \text{MLP}^{(l)}(\text{Norm}(\boldsymbol{h}_{i,\text{post-attn}}^{(l)}))$$

This layered processing allows the model to construct increasingly sophisticated representations from the input, and the $\boldsymbol{h} \in \mathbb{R}^{d_{\text{model}}}$ values are collectively referred to as *activations*. Finally, the output activations from the last layer, $\boldsymbol{h}_i^{(L+1)}$, are projected to logit scores over the vocabulary via an unembedding step, $\text{logits}_i = \text{Unembed}(\boldsymbol{h}_i^{(L+1)})$. These logits are then transformed into probability distributions $\mathbf{y}_i$ for the next token using a softmax function.

**Activation Steering Operators.** Two popular approaches are: *Activation addition* [51] modifies an activation $\boldsymbol{h}$ by adding a scaled feature vector: $\boldsymbol{h}' = \boldsymbol{h} + \alpha \hat{\boldsymbol{d}}_{\text{feat}}$, where $\hat{\boldsymbol{d}}_{\text{feat}}$ denoting the unit-normalized feature direction and $\alpha$ controls the strength of the effect; *Directional ablation* [1] removes the feature by projecting the activation onto the orthogonal complement: $\boldsymbol{h}' = \boldsymbol{h} - \hat{\boldsymbol{d}}_{\text{feat}} \hat{\boldsymbol{d}}_{\text{feat}}^\top \boldsymbol{h}$. While effective, these methods offer limited granularity. Addition is sensitive to coefficient tuning, and orthogonalization removes the feature entirely. Recent works introduce conditional steering [19, 21], which applies these edits selectively, but still rely on the same underlying primitives. Our proposed method, *Angular Steering*, generalizes these interventions as rotation in a 2D subspace, offering continuous, interpretable, and norm-preserving control.

**Choice of Activations for Steering.** There are two main options for choosing the representation for steering: the raw activations [1, 58, 51, 20, 2] or the normalized activations [52]. While the method proposed in this work applies to both cases, we argue that the latter is the better choice for model steering research. Section 4.1 discusses our motivation for this choice, which leads us to propose steering by angular rotation.

## 4 Angular Steering

### 4.1 Motivation for Angular Steering

**Rotation is Better for Steering.** Existing activation steering methods that use vector addition [51] require carefully tuned coefficients, which are highly sensitive to layer-specific activation norms. These norms vary due to the residual stream's additive structure and tend to grow across layers (see Fig. 3, also Fig. 10 (left) in Appendix I), making hyperparameter tuning brittle. Orthogonalization [1] offers a hyperparameter-free alternative but ignores the effects of negative scaling, which prior work suggests can induce opposite behaviors [51, 58, 49].

Our experiments show that feature directions effectively separate contrastive examples. Particularly, in Fig. 4 (also Fig. 10 (right) in Appendix I), for each layer $i$, we plot the scalar projection of the normalized activation $\hat{\boldsymbol{h}}^i$ on the locally extracted feature direction $\boldsymbol{d}_{\text{feature}}^i$ and demonstrate that activations from contrastive datasets aligned oppositely with the local refusal directions.

Furthermore, modern LLMs such as LLAMA 3 [23], QWEN 2.5 [54], and GEMMA 2 [14] use RMSNorm [57] before each MLP and Self Attention block. It is formulated as $\bar{\boldsymbol{h}} = \boldsymbol{h}/\text{RMS}(\boldsymbol{h}) \odot \boldsymbol{g}$, where $\text{RMS}(\boldsymbol{h}) = \sqrt{(1/d_{\text{model}}) \sum_{i=1}^{d_{\text{model}}} \boldsymbol{h}_i^2}$ and $\odot$ denotes element-wise multiplication. This operation first maps the activation to a $\sqrt{d_{\text{model}}}$-scaled unit sphere, making any prior modification

---

[1]Some model families (e.g. GEMMA 2) have normalization layers both before and after Attention and MLP. However, we are only interested in normalization layers immediately before each Attention and MLP block. We also omit other details such as positional embeddings.

effectively norm-preserving, and then directionally scales it into a fixed hyperellipsoid via $\odot \boldsymbol{g}$. Even after rescaling, the activation norms have been shown to remain stable [57].

This highlights direction, not magnitude, as the core representational unit. It also aligns with recent interpretability work supporting the Superposition Hypothesis [10]: that features correspond to nearly orthogonal directions and activations are linear combinations of them [1, 2, 4, 6, 10, 12, 25, 52, 49, 3, 26, 39, 50]. Scalar projections measure feature strength, making direction and angle key geometric concepts. Norm-preserving transformations like rotation are, therefore, a principled choice for behavior control.

**Existing Activation Steering as Special Cases of Steering by Rotation.** Vector arithmetic and orthogonalization using the pre-normalized activation $\boldsymbol{h}^i$ at layer $i$ and a direction representing some feature ($\boldsymbol{d}_{\text{feat}}$) are equivalent to rotation inside a 2D subspace spanned by $Span\{\boldsymbol{h}^i, \boldsymbol{d}_{\text{feat}}\}$ (Fig. 1). When the activation norms are fixed as recommended by [52, 35], existing steering techniques are special cases of angular steering, albeit with restricted flexibility: vector addition is limited to less than 180 degrees, and orthogonalization is fixed at 90 degrees. We provide detailed derivations in Appendix A and empirical comparisons in Appendix C.

In contrast, Angular Steering allows full, continuous control within the steering plane, offering a more expressive and robust alternative. This is further supported by [52], who show that using normalized activations improves probing accuracy across classifiers, reinforcing our hypothesis that steering direction, not raw magnitude, is what ultimately matters.

### 4.2 Overview of Angular Steering

We propose to formulate activation steering as a rotation on a 2-dimensional (2D) subspace $P$ and around the $(d_{\text{model}} - 2)$-dimensional orthogonal complement $Q$ of $P$. Ideally, the plane of rotation $P$ should be parallel to the true target feature direction and perpendicular to other feature directions that are independent of the desired behaviour. Our angular steering provides the following advantages:

- **Generalization.** It is a generalization of existing steering operations (Fig. 1), namely activation arithmetic [51, 58, 2, 39] and directional ablation [1, 58].

- **Universality.** It can be applied to both raw and normalized activations, although the latter is more computationally efficient.

- **Stability.** Restricting the rotation to a 2D subspace confines changes to just two orthogonal directions, leaving the remaining basis vectors unaffected. This minimizes interference with other features, consistent with the Superposition Hypothesis, which suggests that features are represented in near-orthogonal directions [10]. Consequently, this approach enables more robust control over the steering effect, preserving coherence (see Section 5).

- **Flexibility.** It enables steering the activations for more than 180 degrees, making the accuracy less dependent on the quality of the direction of the extracted features.

### 4.3 Preparing Dataset and Models

**Datasets.** To calibrate the feature (refusal) direction, we construct two datasets: $\mathcal{D}_{\text{harmful}}^{(\text{cal})}$, which is a split (80%) of the ADVBENCH dataset [59] consisting of 416 harmful instructions; and $\mathcal{D}_{\text{harmful}}^{(\text{cal})}$, a random subset of 512 harmless examples from the ALPACA dataset [48]. For evaluating steering effectiveness, we use the remaining 20% of ADVBENCH, denoted as $\mathcal{D}_{\text{harmful}}^{(\text{eval})}$, containing 104 samples. To assess general language modeling capabilities, we employ the TINYBENCHMARKS dataset [24], a collection of reduced-scale benchmarks each containing 100 examples: ARC [8], MMLU [15], WINOGRANDE [40], GSM8K [9], TRUTHFULQA [22], and HELLASWAG [56].

**Models.** We show experimental results on steering the refusal feature on various model families (LLAMA 3 [23], QWEN 2.5 [54], GEMMA 2 [14]) of various sizes (3B to 14B). A full list of models used in this work is presented in Appendix D.

### 4.4 Computing the target feature direction

**Extracting Activation Vectors.** Following [1], we pass $\mathcal{D}_{\text{harmful}}^{(\text{cal})}$ and $\mathcal{D}_{\text{harmless}}^{(\text{cal})}$ through the model and record the activations of the final input token after the normalization layers in each transformer block as recommended by [52]. Note that in each transformer block, there are two normalization layers: before the Attention and before the MLP. As a result, we record the activations at two extraction points per transformer block.

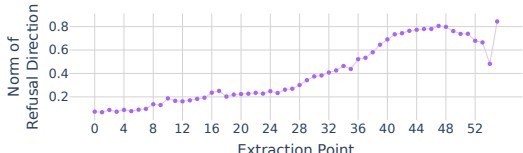
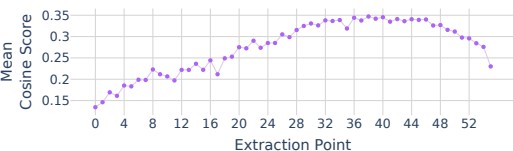

(a) Norm of candidate feature direction at each layer.  (b) Mean cosine similarity of each candidate feature direction with those from other layers.

Figure 5: Statistics of refusal direction candidate for QWEN2.5-7B-INSTRUCT.

**Calculating Candidate Directions.** At each extraction point $i$, we compute a candidate direction using the Difference-in-Means method [3]: $\boldsymbol{d}_{\text{feat}}^i = \overline{\boldsymbol{h}}_{\text{harmful}}^{(\text{cal}),i} - \overline{\boldsymbol{h}}_{\text{harmless}}^{(\text{cal}),i}$ $(i = 1, \ldots, M)$, where $\boldsymbol{d}_{\text{feat}}^i$ is the direction at extraction point $i$, and $\overline{\boldsymbol{h}}_{\text{harmful}}^{(\text{cal}),i}$ and $^i\overline{\boldsymbol{h}}_{\text{harmless}}^{(\text{cal}),i}$ are the means computed over activations from $\mathcal{D}_{\text{harmful}}^{(\text{cal})}$ and $\mathcal{D}_{\text{harmless}}^{(\text{cal})}$, respectively. Here, $M$ is the number of extraction points, defined as twice the number of Transformer blocks in the model. One candidate direction is computed at each extraction point, yielding a total of $M$ candidate directions.

**Choosing One Feature Direction.** Among $M$ candidate directions, we choose a feature direction for Angular Steering. Fig. 5b shows high cosine similarity among candidate directions in layers where refusal is strong, suggesting those directions are stable approximations of the true feature. This observation suggests that the similarity between candidate directions can be a promising metric to select the feature direction. In Angular Steering, we choose the candidate direction $\hat{\boldsymbol{d}}_{\text{feat}}$ that is most similar to others as the feature direction. We normalize $\hat{\boldsymbol{d}}_{\text{feat}}$ to make it a unit vector.

**Remark 1 (Automatic Direction Selection)** *Unlike [1], which selects directions manually, we use a simple statistical procedure to choose the feature direction automatically. Though hand-tuning might yield better downstream results, we aim to study steering control rather than maximize performance.*

**Remark 2** *Fig. 4 and Fig. 5 shows that refusal behavior emerges progressively along the depth of the model, stabilizes, and then spikes again near the final layer. We hypothesize that this late spike reflects a filtering step just before token generation and thus omit this point from the list of candidates.*

### 4.5 Selecting the Steering Plane

We now require a second direction to define the 2D steering plane in Angular Steering. As discussed in Section 4.1, the optimal plane should maximize the influence on the feature of interest while minimizing unintended impacts on other features. While using the $Span\{\boldsymbol{h}^i, \hat{\boldsymbol{d}}_{\text{feat}}\}$ aligns with prior methods like directional ablation and activation addition, we argue against it due to three reasons: (1) prior work suggests that feature directions are layer-independent [33, 10, 50, 1], implying a shared geometry across layers; (2) this span might include other dominant features, risking general degradation [51, 49]; and (3) computing rotation at each step is costly. Instead, we propose a fixed plane that isolates the feature of interest.

To construct this fixed plane, we perform PCA on the candidate directions $\boldsymbol{d}_{\text{feat}}^i$ and select the first principal component, $\hat{\boldsymbol{d}}_{\text{PC0}}$, as the second axis. This captures variance across layers, which, as shown in prior work [1, 52, 20, 58], reflects variation in approximating the true feature direction. The resulting plane $Span(\hat{\boldsymbol{d}}_{\text{feat}}, \hat{\boldsymbol{d}}_{\text{PC0}})$ thus isolates meaningful variation in the target feature. Fig. 6 shows a smooth directional shift across layers in this plane, supporting the hypothesis that feature strength evolves gradually, making it a natural basis for steering (see Section 5).

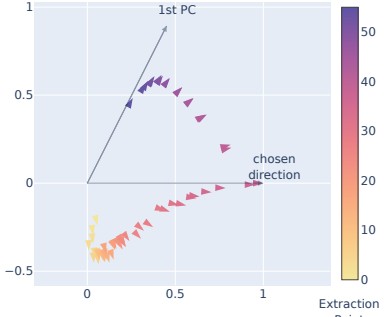

Figure 6: Projections of the feature directions extracted at each layer (i.e. $\boldsymbol{d}_{\text{feat}}^i$) on the steering plane for QWEN2.5-7B-INSTRUCT.

### 4.6 Putting It All Together:
### The (Adaptive) Angular Steering Framework

We are now ready to formulate Angular Steering and its adaptive variant.

#### 4.6.1 Angular Steering Framework

Let $P$ be the 2D subspace spanned by $\hat{d}_{\text{feat}}$ and $\hat{d}_{\text{PC0}}$. We compute the orthonormal basis $\{b_1, b_2\}$ of $P$ as follows:

$$b_1 \leftarrow \hat{d}_{\text{feat}}; \quad b_2 \leftarrow \hat{d}_{\text{PC0}} - (\hat{d}_{\text{PC0}} \cdot b_1)b_1; \quad b_2 \leftarrow b_2 / ||b_2||.$$

**Rotation by an Offset Angle.** To rotate within the subspace $P$ by an angle $\phi$, the transformation matrix $R_\phi^P$ is given as

$$R_\phi^P = I - (b_1 b_1^\top + b_2 b_2^\top) + [b_1 \ b_2] R_\phi [b_1 \ b_2]^\top \tag{1}$$

where $I - (b_1 b_1^\top + b_2 b_2^\top)$ is the projection to the $(d_{\text{model}} - 2)$-dimensional orthogonal complement $Q$ of $P$ and $R_\phi$ is the 2D rotation matrix given as $R_\phi = \begin{bmatrix} cos(\phi) & -sin(\phi) \\ sin(\phi) & cos(\phi) \end{bmatrix}$.

**Rotation to a Target Angle.** In practice, rather than rotating all activations by a fixed offset, we often want to rotate them to a specific angular position $\theta$, e.g., where a desired behaviour is strongly expressed. A naive approach would involve: (1) projecting the input $h$ onto the steering plane $P$: $\text{proj}_P(h) = (b_1 b_1^\top + b_2 b_2^\top) \cdot h$; (2) computing the current angle $\phi_{h,b_1}^P$ between $\text{proj}_P(h)$ and $b_1$; (3) constructing the rotation matrix $R_{\theta-\phi}^P$ using Eqn. 1; and (4) applying this matrix to $h$. However, this is inefficient when $\theta$ is fixed and can be optimized by precomputing reusable components.

Noting that the term $[b_1 \ b_2] R_\phi [b_1 \ b_2]^\top$ in Eqn. 1 is a norm-preserving transformation, we can precompute its effect on the unit vector $[1 \ 0]^\top$ and scale the result by $|\text{proj}_P(h)|$. This leads to the following efficient formulation for rotating an input $h$ to angle $\theta$:

$$h_{\text{steered},\theta} = R_{\theta-\phi_{h,b_1}}^P \cdot h = h - \text{proj}_P(h) + |\text{proj}_P(h)| \cdot [b_1 \ b_2] R_\theta [1 \ 0]^\top, \tag{2}$$

where $R_{\theta-\phi_{h,b_1}}^P$ is the rotation matrix defined in Eqn. 1. Here, both the projection matrix $(b_1 b_1^\top + b_2 b_2^\top)$ and $[b_1 \ b_2] R_\theta [1 \ 0]^\top$ can be precomputed.

#### 4.6.2 Adaptive Angular Steering Framework

Since inputs from contrastive datasets tend to align with $d_{\text{feat}}^i$ in opposite directions (Fig. 4), it is unnecessary to rotate all activations uniformly. To increase flexibility and further reduce unintended effects on non-targeted features, we propose an adaptive variant that rotates only activations positively aligned with $\hat{d}_{\text{feat}}$. In particular, we first compute a conditional mask based on the sign of the projection onto $\hat{d}_{\text{feat}}$: $\text{mask} = \max(0, \ \text{sign}(\text{proj}_{\hat{d}_{\text{feat}}}(x)))$. Using this mask, Eqn. 2 becomes:

$$h_{\text{steered (adaptive)},\theta} = h + \text{mask} \cdot \left( |\text{proj}_P(h)| \cdot [b_1 \ b_2] R_\theta [1 \ 0]^\top - \text{proj}_P(h) \right) \tag{3}$$

This formulation adds an additional layer of control and robustness: steering is both restricted to a 2D subspace and selectively applied based on feature alignment. Beyond adjusting the steering angle $\theta$, users may also vary the similarity threshold used in the mask or employ different $d_{\text{feat}}^i$ across layers. We note that another conditional steering approach has been explored in contemporary work by [19], but activation addition was used as the steering framework instead of rotation. We summarize the algorithms for feature direction extraction, steering plane selection, and angular steering in Appendix B.

## 5 Controlling the Steering Effect

For inference, we apply Adaptive Angular Steering as described in Eqn. 3 on every normalization module before each Attention and MLP layer. By varying the target angular position $\theta$ from 0 to 360 degrees (with 10-degree intervals), we observe that the models change from refusal to compliance and back to refusal again (see Fig. 7). We found that both Angular Steering and Adaptive Angular Steering are effective at varying the steering effect. However, the non-adaptive version runs a risk of breaking the coherence on smaller models, which will be discussed in Section 6.

**Remark 3** *In addition to the evaluation of refusal steering presented in this section, we also assess our method's ability to control various emotions, with results reported in Appendix H.*

**Evaluation Metrics.** We compute a **refusal score** using the substring matching method [1], which operates by matching a set of common "refusal substrings" (e.g., "I'm sorry", "As an

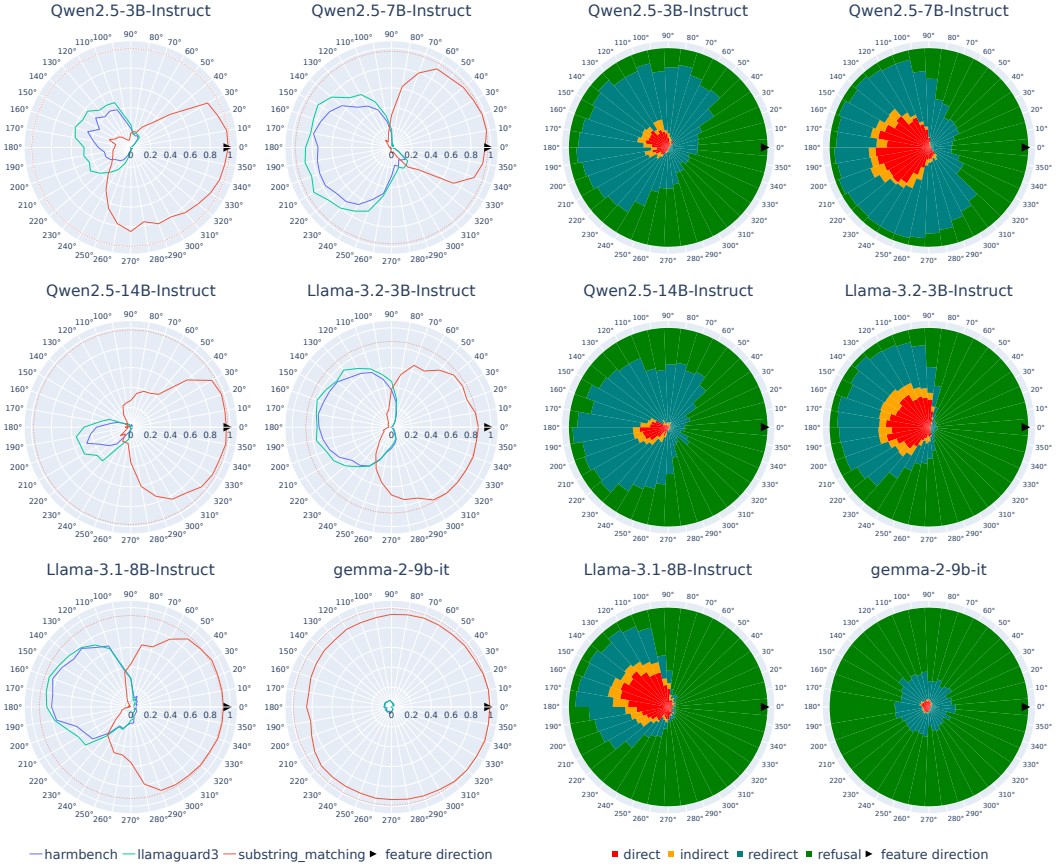

(a) Refusal score (substring matching [1]) and harmful scores (LLAMAGUARD3 [23], HARMBENCH [27]).

(b) LLM-as-a-judge classification results: models' responses are classified by an LLM into four categories: direct, indirect, redirect and refusal.

Figure 7: **Steering evaluation.** Each model was steered using Adaptive Angular Steering and evaluated at every 10-degree angular position along the steering circle. Solid traces show evaluation scores; dashed traces indicate baseline (non-steered) models. Traces of the same color correspond to the same benchmark. Baseline values for LLAMAGUARD3 and HARMBENCH may be hidden due to near-zero values.

AI") on the model completion. The score is 1 if at least one such substring is matched and 0 otherwise.

Intuitively, this metric only detects memorized refusal phrases but does not assess coherence and harmfulness, as noted by [1, 16, 29, 36, 41]. To evaluate harmfulness, we follow the setup in [1] and use two more complementary evaluation metrics, LLAMAGUARD3 [23] and HARMBENCH [27], which we collectively call **harmful scores**. These two methods use open-source models to classify whether an input is harmful, in which the score is 1 if the classification is true and 0 otherwise.

Beyond refusal and harmfulness detection, we are interested in how the model's output changes semantically at different level of refusal. Thus, we perform qualitative analysis using a reasoning model QVQ-72B-PREVIEW [38] to classify the generation outputs into 4 classes: direct: The model directly answers the prompt; indirect: The model starts out seemingly unwilling to answer but then still provides with an answer; redirect: The model does not explicitly agree or refuse to answer but provides a tactful response without producing any harmful content; refusal: The model explicitly refuses to answer.

**Evaluation along the Steering Circle.** Fig.7 demonstrates that angular steering effectively modulates refusal and safety behaviors. In Fig.7a, all models show a clear arc of strong alignment–high refusal and low harmful scores–and an opposing arc of weak alignment–low refusal and high harmful scores. These arcs lie in opposite directions within the steering circle, with performance peaking near the center and diminishing outward. Fig.7b further supports this observation by showing that, for five of six models, refusal dominates in the strong arc, followed by redirect, and then

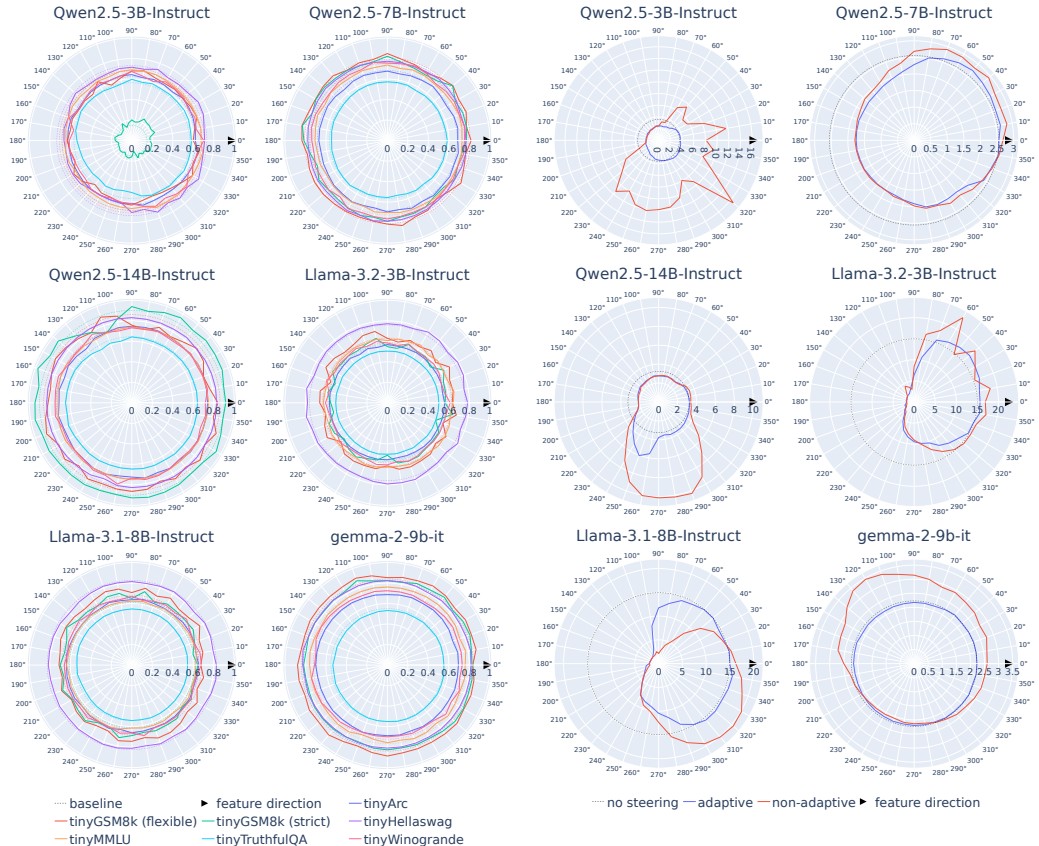

(a) Benchmark results on the TINYBENCHMARKS [24] suite.

(b) Perplexity scores of generations from Adaptive Steering, non-adaptive Steering and no steering.

Figure 8: **Evaluation beyond the targeted steering task.** Each model was steered using Adaptive Angular Steering (Eqn. 3) and evaluated on all benchmarks at every 10-degree angular position along the steering circle. Solid traces represent evaluation scores along the steering circle, and dashed traces represent the evaluation for the baseline (non-steered models); traces having the same color represent the same benchmark.

direct or indirect responses as the angle shifts. Tab.1 reports example completions for each class. GEMMA-2-9B-IT is an exception, displaying the weakest effect yet still following the overall trend.

**Steering on a random plane.** For completeness, we conduct an ablation study on steering using Adaptive Angular Steering with a random plane. Fig. 14b in Appendix I.2 shows that it has little to no effect on controlling refusal in five out of six tested models.

## 6 Effects on Model's Performance beyond the Targeted Steering Task

Steering can degrade language modeling ability [43], especially when relying on sensitive hyperparameters [51, 58, 49, 2, 20, 52], which may lead to incoherent outputs if not carefully tuned [51, 49]. In this section, we quantitatively assess the impact of our method on overall LLM performance.

### 6.1 Language Modeling Benchmarks

**Method.** For each model, we adaptively steer its activation with a $10°$ interval along the entire steering circle using Eqn. 3 and evaluate all benchmarks from the TINYBENCHMARKS suite [24]. The results are visualized in Fig. 8a.

**Results.** Overall, our steering method effectively preserves benchmark accuracies across the entire steering circle, demonstrating strong robustness. Interestingly, in many cases, performance under intervention even surpasses the non-steered baseline.

A notable outlier is QWEN2.5-3B-INSTRUCT, which exhibits a performance drop along the arc from $160°$ to $280°$. We attribute this to feature interference [10], where multiple latent features dominate

within the chosen steering plane, a phenomenon to which smaller models are more susceptible. The consistent accuracy drop across all benchmarks in this region suggests the model is reacting to a competing feature. For TINYGSM8K, although the model often generates a correct answer, it fails to match the expected format, leading to significantly lower scores under the `strict` metric compared to the more lenient `flexible` variant.

It is important to note that for TINYGSM8K, the `flexible` metric extracts the last numeric value as the final answer, whereas the `strict` variant assumes a predefined output format. Consequently, these metrics are highly sensitive to formatting variations, leading to noticeable fluctuations in accuracy across different steering angles.

## 6.2 Perplexity of the Steered Generations

**Smaller Models are More Vulnerable to Interference under Angular Steering.** In non-adaptive Angular Steering experiments, 7B-14B models generate coherent outputs throughout the steering circle, while smaller models like LLAMA-3.2-3B-INSTRUCT and QWEN2.5-3B-INSTRUCT often produce incoherent text across a wide arc. Notably, refusal phrases still appear randomly in various languages for LLAMA-3.2-3B-INSTRUCT, and mainly in Chinese for QWEN2.5-3B-INSTRUCT, despite English prompts. This suggests that limited capacity in smaller models leads to feature interference [10], with multiple features entangled in the 2D steering subspace, as discussed in Sections 5 and 6.1.

**Method.** Motivated by such observations, we analyze the perplexity of the steered generations using the non-steered models and report the results in Fig. 8b. Given an input sequence $x$, an non-steered LLM $\pi_{\text{non-steered}}$, the output is modeled by $y_{\text{non-steered}} \sim \pi_{\text{non-steered}}(x)$. Similarly, $\pi_{\text{steered}}$ and $y_{\text{steered}}$ denote the steered model and its output, respectively. We denote the perplexity score of $x$ with respect to a model $\pi$ as $PPL_\pi(x)$. In Fig. 8b, we compare $PPL_{\pi_{\text{non-steered}}}(x||y_{\text{non-steered}})$, $PPL_{\pi_{\text{non-steered}}}(x||y_{\text{steered (non-adaptive)}})$ and $PPL_{\pi_{\text{non-steered}}}(x||y_{\text{steered (adaptive)}})$ for each model and at every $10°$ rotation.

**Results.** *Both 3B models exhibit unstable perplexity under non-adaptive steering, indicating vulnerability to interference.* For QWEN2.5-3B-INSTRUCT, perplexity remains significantly above baseline across more than half of the circle, aligning with the incoherent outputs discussed earlier. In contrast, LLAMA-3.2-3B-INSTRUCT shows perplexity closer to baseline, consistent with its behavior of still refusing harmful requests, albeit in different languages.

*Adaptive Steering effectively preserves coherence.* Fig. 8b reveals that the perplexity of Adaptive Steering is lower, more stable, and closer to no steering than its non-adaptive counterpart, indicating effectiveness at balancing behavior control with coherence and performance.

*Alignment masks rather than removes harmful behavior.* Perplexity stays near baseline when steering aligns with the target feature, but drops below baseline as it moves toward the jailbroken region. Our perplexity analysis was inspired by [6], which shows that safety alignment mainly affects the first few tokens, while the probability of later harmful tokens remains largely unchanged, suggesting a shortcut where models shift early output distributions without removing harmful behaviors. Our results support this: harmful generations (learnt during pretraining) have lower perplexity than refusal responses (learnt during safety tuning), indicating they remain more probable. While the mechanisms behind safety alignment are still unclear, our findings offer a glimpse into this issue.

## 7 Concluding Remarks

We propose Angular Steering, a novel activation steering method offering continuous, fine-grained control over large language model behaviors by rotating activation vectors within a two-dimensional subspace. This geometric perspective unifies prior steering techniques, enhancing interpretability and deepening understanding of model mechanisms without compromising general performance. Our adaptive variant further improves robustness by selectively applying steering based on context. A limitation of Angular Steering is that while promising, it currently relies on heuristically selected steering planes, which might not always generalize optimally across diverse behaviors or architectures. Future work should focus on systematically identifying effective subspaces and extending adaptive strategies to support broader alignment goals.

## Acknowledgments and Disclosure of Funding

This research / project is supported by the National Research Foundation Singapore under the AI Singapore Programme (AISG Award No: AISG2-TC-2023-012-SGIL). This research / project is supported by the Ministry of Education, Singapore, under the Academic Research Fund Tier 1 (FY2023) (A-8002040-00-00, A-8002039-00-00). This research / project is also supported by the NUS Presidential Young Professorship Award (A-0009807-01-00) and the NUS Artificial Intelligence Institute–Seed Funding (A-8003062-00-00).

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

# Supplement to "Angular Steering: Behavior Control via Rotation in Activation Space"

**Table of Contents**

# A  Detailed Derivation: Existing Activation Steering as Special Cases of Steering by Rotation

We will show that, when activation norms are preserved, existing steering techniques are special cases of angular steering but with restricted flexibility: vector addition is limited to less than 180 degrees, and orthogonalization is fixed at 90 degrees.

Formally, let the activation $h_i$ be decomposed into components parallel and orthogonal to a unit-norm feature direction $\hat{d}_{\text{feat}}$ (for brevity, here we denote them as $h$ and $\mathbf{d}$ respectively):

$$h = (h \cdot d)d + h_\perp, \quad \text{where} \quad h_\perp = h - (h \cdot d)d.$$

Let $\mathbf{u} = \frac{h_\perp}{\|h_\perp\|}$, and define the initial angle between $h$ and $d$ as:

$$\theta_0 = \tan^{-1}\left(\frac{\|h_\perp\|}{h \cdot d}\right).$$

We define *Angular Steering* as rotating $h$ by an offset angle $\phi$ in the plane $Span\{h, d\}$, producing a vector:

$$h_{\text{rot}}(\phi) = \cos(\theta_0 + \phi) \cdot d + \sin(\theta_0 + \phi) \cdot \mathbf{u}.$$

Now consider *vector addition* [51], defined as:

$$h_{\text{add}} = h + \alpha d = (h \cdot d + \alpha)d + h_\perp.$$

After normalization, the direction becomes:

$$h_{\text{add-norm}} = \frac{h_{\text{add}}}{\|h_{\text{add}}\|} = \cos(\theta_0 + \phi_{\text{add}}) \cdot d + \sin(\theta_0 + \phi_{\text{add}}) \cdot \mathbf{u},$$

where $\phi_{\text{add}} = \tan^{-1}\left(\frac{\|h_\perp\|}{h \cdot d + \alpha}\right) - \theta_0$.

Likewise, *directional ablation (orthogonalization)* [1], given by:

$$h_{\text{ablate}} = h_\perp,$$

after normalization becomes:

$$h_{\text{ablate-norm}} = \mathbf{u} = \cos(\theta_0 + \phi_{\text{ablate}}) \cdot d + \sin(\theta_0 + \phi_{\text{ablate}}) \cdot \mathbf{u},$$

with $\phi_{\text{ablate}} = \frac{\pi}{2} - \theta_0$.

Thus, when activation norms are fixed, both addition and ablation shift the direction of $h$ in a way that is exactly equivalent to a rotation by some angle $\phi$ within the plane spanned by $h$ and $d$. This establishes them as special cases of Angular Steering.

In practice, RMSNorm [57] stabilizes activation norms rather than fixing them to exact values (as shown in [57] and Fig. 10, left). As a result, these methods can still be interpreted as approximate rotations.

# B  Algorithms for Angular Steering

---

**Algorithm 1** Extract Feature Direction

---

**Require:** Contrastive datasets $\mathcal{D}_{\text{harmful}}$, $\mathcal{D}_{\text{harmless}}$, model $\mathcal{M}$
1: **for** each layer $i$ in model **do**
2:    Compute normalized activations $\boldsymbol{h}^{(i)}$ after Attention and MLP
3:    Compute mean activation for each dataset:

$$\bar{\boldsymbol{h}}^{(i)}_{\text{harmful}}, \ \bar{\boldsymbol{h}}^{(i)}_{\text{harmless}}$$

4:    Compute candidate direction:

$$\boldsymbol{d}^{(i)} = \bar{\boldsymbol{h}}^{(i)}_{\text{harmful}} - \bar{\boldsymbol{h}}^{(i)}_{\text{harmless}}$$

5: **end for**
6: Select final feature direction $\boldsymbol{d}$ using max average cosine similarity:

$$\boldsymbol{d} = \operatorname*{argmax}_{i=1\dots|\text{layers}|} \left( \frac{1}{|\text{layers}|} \sum_{j=1}^{|\text{layers}|} cosine(\boldsymbol{d}^{(i)}, \boldsymbol{d}^{(j)}) \right)$$

7: Normalize: $\hat{\boldsymbol{d}} = \frac{\boldsymbol{d}}{\|\boldsymbol{d}\|}$

---

**Algorithm 2** Select Steering Plane

---

**Require:** Candidate directions $\{\boldsymbol{d}^{(i)}\}$, feature direction $\hat{\boldsymbol{d}}$
1: Perform PCA on $\{\boldsymbol{d}^{(i)}\}$
2: Let first principal component be $\boldsymbol{d}_{\text{1stPC}}$
3: Set orthonormal basis for plane:

$$\boldsymbol{b}_1 \leftarrow \hat{\boldsymbol{d}}, \quad \boldsymbol{b}_2 \leftarrow \boldsymbol{d}_{\text{1stPC}} - (\boldsymbol{b}_1 \cdot \boldsymbol{d}_{\text{1stPC}})\boldsymbol{b}_1; \quad \boldsymbol{b}_2 \leftarrow \frac{\boldsymbol{b}_2}{\|\boldsymbol{b}_2\|}$$

4: Define projection matrix $P = \boldsymbol{b}_1\boldsymbol{b}_1^\top + \boldsymbol{b}_2\boldsymbol{b}_2^\top$

---

**Algorithm 3** Angular Steering (with optional Adaptive Mask)

---

**Require:** Activation $\boldsymbol{h}$, basis $\boldsymbol{b}_1, \boldsymbol{b}_2$, target angle $\theta$, (optional) mask flag
1: Project: $\text{proj}_P(\boldsymbol{h}) = P \cdot \boldsymbol{h}$
2: Compute magnitude: $\boldsymbol{r} = \|\text{proj}_P(\boldsymbol{h})\|$
3: Precompute: $\boldsymbol{v}_\theta = [\boldsymbol{b}_1 \ \boldsymbol{b}_2] \cdot R_\theta \cdot [1 \ 0]^\top$
4: **if** adaptive **then**
5:    Compute mask: $\text{mask} = \max(0, \text{sign}(\boldsymbol{h} \cdot \hat{\boldsymbol{d}}))$
6:    Apply adaptive steering:

$$\boldsymbol{h}_{\text{steered}} = \boldsymbol{h} + \text{mask} \cdot (\boldsymbol{r} \cdot \boldsymbol{v}_\theta - \text{proj}_P(\boldsymbol{h}))$$

7: **else**
8:    Apply steering:
$$\boldsymbol{h}_{\text{steered}} = \boldsymbol{h} - \text{proj}_P(\boldsymbol{h}) + \boldsymbol{r} \cdot \boldsymbol{v}_\theta$$

9: **end if**

---

# C  Comparison with existing methods

## C.1  Steering Performance Comparison

In Tab. 2, we compare refusal steering performance between our method, prior approaches, and the no-steering baseline. To ensure a fair and consistent setup, we employ the protocol below:

- Following observations in [1, 32, 28] that multi-layer interventions yield better results, we apply steering across all layers for methods considered in this study.
- All methods perform steering within the subspace $Span(\boldsymbol{h}, \boldsymbol{d}_{\text{feat}})$, as in [1, 51].
- We conduct hyperparameter tuning for both Angular Steering and Activation Addition. For Activation Addition, tuning is notably *more complex and time-consuming*, requiring layer-wise unbounded coefficients. In contrast, our method only uses a single bounded rotation angle.

**Results:** Across all evaluated models, our method *achieves equal or better refusal performance than existing methods*, supporting our theoretical insights.

Table 2: Comparison of refusal steering benchmarks across models. Arrows indicate the desired direction of change.

| Model | No Steering | AAS (Ours) | Addition | Ablation |
|---|---|---|---|---|
| QWEN2.5-3B-INSTRUCT | | | | |
| HARMBENCH ↑ | 0.0000 | 0.6635 | 0.6635 | 0.2019 |
| LLAMAGUARD3 ↑ | 0.0000 | 0.9904 | 0.9904 | 0.2212 |
| substring matching ↓ | 1.0000 | 0.0000 | 0.0000 | 0.1731 |
| QWEN2.5-7B-INSTRUCT | | | | |
| HARMBENCH ↑ | 0.0192 | 0.8750 | 0.8750 | 0.3942 |
| LLAMAGUARD3 ↑ | 0.0000 | 1.0000 | 0.9808 | 0.5288 |
| substring matching ↓ | 0.9712 | 0.0000 | 0.0000 | 0.0577 |
| QWEN2.5-14B-INSTRUCT | | | | |
| HARMBENCH ↑ | 0.0192 | 0.7212 | 0.7212 | 0.0288 |
| LLAMAGUARD3 ↑ | 0.0000 | 1.0000 | 0.9904 | 0.0385 |
| substring matching ↓ | 0.9808 | 0.0000 | 0.0000 | 0.0962 |
| LLAMA-3.2-3B-INSTRUCT | | | | |
| HARMBENCH ↑ | 0.0096 | 0.7212 | 0.7212 | 0.1058 |
| LLAMAGUARD3 ↑ | 0.0000 | 0.9904 | 0.9904 | 0.0865 |
| substring matching ↓ | 0.7885 | 0.0000 | 0.0000 | 0.7885 |
| LLAMA-3.1-8B-INSTRUCT | | | | |
| HARMBENCH ↑ | 0.0577 | 0.8173 | 0.8173 | 0.0577 |
| LLAMAGUARD3 ↑ | 0.0385 | 0.9904 | 0.9904 | 0.0385 |
| substring matching ↓ | 0.9231 | 0.0000 | 0.0000 | 0.9231 |
| GEMMA-2-9B-IT | | | | |
| HARMBENCH ↑ | 0.0096 | 0.6250 | 0.6250 | 0.0000 |
| LLAMAGUARD3 ↑ | 0.0000 | 0.9904 | 0.9904 | 0.0000 |
| substring matching ↓ | 0.9904 | 0.0000 | 0.0000 | 1.0000 |

## C.2  Robustness Comparison

In Tab. 3 and Fig. 9, we examine the models coherence and general performance under two different steering subspaces:

- $Span(\boldsymbol{h}, \boldsymbol{d}_{\text{feat}})$ (used in [1, 51])
- $Span(\hat{\boldsymbol{d}}_{\text{PC0}}, \boldsymbol{d}_{\text{feat}})$ (our proposal)

### C.2.1 Perplexity Analysis

Similar to Fig. 8b, Fig. 9a show comparisons of perplexity scores between steering within $Span(d_{\text{feat}}, \hat{d}_{\text{PC0}})$ (ours) and within $Span(h, d_{\text{feat}})$ (used by exsting work).

In Tab. 3, we report the following metrics: *mean*, *max*, *min* perplexity values across different steering angles; and *mean diff* the average difference in perplexity between consecutive angles, which indicates the model's sensitivity to small hyperparameter changes.

**Results:** Steering on $Span(\hat{d}_{\text{PC0}}, d_{\text{feat}})$ yields low and stable perplexity, demonstrating strong coherence across steering angles. In contrast, steering on $Span(h, d_{\text{feat}})$ causes larger fluctuations and higher perplexity, indicating greater sensitivity and frequent coherence breakdowns (e.g., generating gibberish), a finding consistent with our qualitative observations.

Table 3: Perplexity comparison across models. Lower perplexity and smaller mean difference indicate more stable and coherent generation under steering.

| Metric | Baseline | $Span(\hat{d}_{\text{PC0}}, d_{\text{feat}})$ (Ours) | $Span(h, d_{\text{feat}})$ |
|---|---|---|---|
| QWEN2.5-3B-INSTRUCT | | | |
| *mean* | 3.5772 | 2.9303 | 5.6141 |
| *max* | 3.5772 | 4.0295 | 56.7403 |
| *min* | 3.5772 | 2.1080 | 1.5398 |
| *mean diff* | 0.0000 | 0.1201 | 6.9214 |
| QWEN2.5-7B-INSTRUCT | | | |
| *mean* | 2.5554 | 2.1168 | 4.8154 |
| *max* | 2.5554 | 2.7457 | 33.4639 |
| *min* | 2.5554 | 1.7167 | 1.4330 |
| *mean diff* | 0.0000 | 0.0643 | 2.3969 |
| QWEN2.5-14B-INSTRUCT | | | |
| *mean* | 3.2461 | 3.2056 | 3.6165 |
| *max* | 3.2461 | 6.0337 | 12.9603 |
| *min* | 3.2461 | 2.1199 | 1.5721 |
| *mean diff* | 0.0000 | 0.2372 | 1.3552 |
| LLAMA-3.2-3B-INSTRUCT | | | |
| *mean* | 14.9902 | 8.7736 | 3.7316 |
| *max* | 14.9902 | 17.1567 | 33.7329 |
| *min* | 14.9902 | 1.7603 | 1.6163 |
| *mean diff* | 0.0000 | 0.8891 | 2.8426 |
| LLAMA-3.1-8B-INSTRUCT | | | |
| *mean* | 14.9360 | 9.3937 | 15.2867 |
| *max* | 14.9360 | 15.7313 | 62.1794 |
| *min* | 14.9360 | 1.7601 | 1.5726 |
| *mean diff* | 0.0000 | 0.8215 | 12.2612 |
| GEMMA-2-9B-IT | | | |
| *mean* | 2.2298 | 2.1850 | 8.3022 |
| *max* | 2.2298 | 2.2541 | 35.0758 |
| *min* | 2.2298 | 2.1240 | 1.2172 |
| *mean diff* | 0.0000 | 0.0160 | 4.8397 |

### C.2.2 General Language Modeling

Similar to Fig. 8a, in Fig. 9a, we show general task performance on TINYBENCHMARKS when steering within $Span(h, d_{\text{feat}})$.

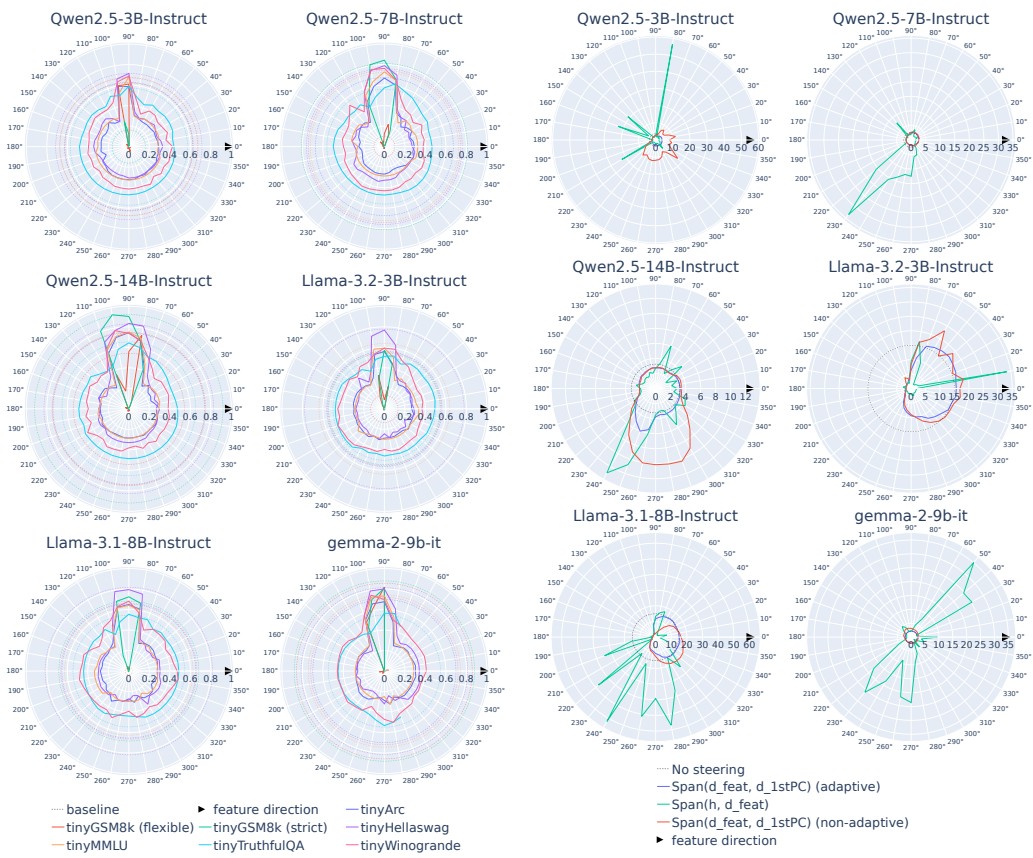

(a) Benchmark results on the TINYBENCHMARKS when steering within $Span(h, d_{feat})$ [24] suite.

(b) Perplexity scores comparison between steering within $Span(d_{feat}, \hat{d}_{PC0})$ (ours) and within $Span(h, d_{feat})$ (existing work).

Figure 9: **Robustness comparisons between steering within $Span(d_{feat}, \hat{d}_{PC0})$ (ours) and within $Span(h, d_{feat})$ (existing work).** Each model was steered using Adaptive Angular Steering (Eqn. 3) and evaluated on all benchmarks at every 10-degree angular position along the steering circle. Solid traces represent evaluation scores along the steering circle, and dashed traces represent the evaluation for the baseline (non-steered models); traces having the same color represent the same benchmark.

**Results:** Steering within $Span(\hat{d}_{PC0}, d_{feat})$ preserves performance across most angles. In contrast, steering on $Span(h, d_{feat})$ causes significant performance drops, except near $90°$ where performance temporarily aligns with the baseline, consistent with prior observations [1].

These results support our hypothesis that $Span(h, d_{feat})$ overlaps with unrelated directions, leading to interference. In contrast, our proposed subspace more effectively isolates the target feature, yielding more robust and controllable steering.

# D Use of existing assets

## D.1 Models

Table 4: Models used in this work.

| Model (with link) | Usage | Source | License |
|---|---|---|---|
| QWEN2.5-(3B, 7B, 13B)-INSTRUCT [54] | Experimental subject | HF Hub | Apache license 2.0 |
| LLAMA-3.1-8B-INSTRUCT [23] | Experimental subject | HF Hub | Llama 3.1 Community License Agreement |
| LLAMA-3.2-3B-INSTRUCT [23] | Experimental subject | HF Hub | Llama 3.2 Community License Agreement |
| GEMMA-2-9B-IT [14] | Experimental subject | HF Hub | Gemma Terms of Use |
| LLAMA-GUARD-3-8B [23] | Evaluation device | HF Hub | Llama 3.1 Community License Agreement |
| HARMBENCH CLASSIFIER [27] | Evaluation device | HF Hub | MIT |
| QVQ-72B-PREVIEW [38] | Evaluation device | HF Hub | Qwen License |
| EMOLLAMA-CHAT-13B [55] | Evaluation device | HF Hub | MIT |

## D.2 Datasets

Table 5: Datasets used in this work.

| Dataset (with link) | Source | License |
|---|---|---|
| ADVBENCH [59] | Github | MIT |
| ALPACA [48] | HF Hub | Creative Commons Attribution Non Commercial 4.0 |
| TINYBENCHMARKS [24] | Github | MIT |

# E    Compute statement

This research was conducted using mainly Nvidia H100 GPUs with 80GB of memory. For each model:

- Constructing the steering plane took about 15 minutes on 1 GPU using TRANSFORMER-LENS [30].
- Pre-generating responses for evaluation took about 10 minutes on 1 GPU using our fork of vLLM [18] as the serving engine.
- Evaluation with `substring matching` [1], LLAMA 3 GUARD [23] and HARM-BENCH [27] collectively took about 10 minutes on 1 GPU using vLLM [18] as the serving enging.
- Evaluation with LLM-as-a-judge took about 50 minutes on 4 GPUs using vLLM [18] as the serving engine.
- Computing perplexity scores took about 5 minutes on 1 GPU.
- Evaluation with TINYBENCHMARKS [24] took about 4 hours on 1 GPU using vLLM [18] as the serving engine and LM HARNESS [13] as the evaluation device.

# F    Computational and memory complexity analysis

Overall, our method has a time complexity of $O(|\text{transformer layers}| \times d_{\text{model}}^2)$ and a memory complexity of $O(d_{\text{model}}^2)$ where $d_{\text{model}}$ is the dimension of the transformer layers' hidden states. For each token at each intervention point, (Adaptive) Angular Steering makes two matrix multiplications and a few element-wise operations. In terms of memory, our formulation enables us to pre-compute one $d_{\text{model}} \times d_{\text{model}}$ matrix and one $d_{\text{model}}$-dimensional vector, which are shared across all extraction points. Below we present the detail analysis of the time and memory complexity of our method.

Recall the Adaptive Angular Steering formula is:

$$\text{mask} = \max(0, \ \text{sign}(\text{proj}_{\boldsymbol{d}_{\text{feat}}}(\boldsymbol{h})))$$

$$\mathbf{h}_{\text{steered (adaptive)},\theta} = \boldsymbol{h} + \text{mask} \cdot \left(|\text{proj}_P(\boldsymbol{h})| \cdot [b_1 \, b_2] R_\theta [1 \, 0]^\top - \text{proj}_P(\boldsymbol{h})\right)$$

with

- $\boldsymbol{h} \in \mathbb{R}^{d_{\text{model}}}$: the activation at some intervention point.
- $P$: the 2D rotation subspace.
- $\{b_1, b_2\} \in \mathbb{R}^{d_{\text{model}}}$: the orthonormal basis of $P$.
- $\theta$: the target angular position.
- $R_\theta \in \mathbb{R}^{2 \times 2}$: the 2D rotation matrix to $\theta$.
- $\text{proj}_y(x)$ denotes the projection of $x$ onto $y$.

The formulation above was chosen with the intention that some components can be pre-computed:

- $(\boldsymbol{b}_1 \boldsymbol{b}_1^\top + \boldsymbol{b}_2 \boldsymbol{b}_2^\top) \in \mathbb{R}^{d_{\text{model}} \times d_{\text{model}}}$: the projection matrix for $\text{proj}_P(\cdot)$.
- $[\boldsymbol{b}_1 \, \boldsymbol{b}_2] R_\theta [1 \, 0]^\top \in \mathbb{R}^{d_{\text{model}}}$

Hence the complexity of the above operation is:

**Time** (per token): $O(|\text{transformer layers}| \times d_{\text{model}}^2)$ *(assuming the naive implementation of matrix multiplication)*

- Computing $\text{proj}_P(\boldsymbol{h})$ takes $O(d_{\text{model}}^2)$.
- Computing $\text{proj}_{\boldsymbol{d}_{\text{feat}}}(\boldsymbol{h})$ takes $O(d_{\text{model}})$
- Other element-wise operations $(\text{sign}, \max, \cdot, +, -)$ each takes $O(d_{\text{model}})$.
- The operation is applied at each intervention point and the number of intervention points is $O(|\text{transformer layers}|)$.

**Memory**: $O(d_{\text{model}}^2)$

- Storing $(\boldsymbol{b}_1 \boldsymbol{b}_1^\top + \boldsymbol{b}_2 \boldsymbol{b}_2^\top)$ takes $O(d_{\text{model}}^2)$
- Storing $[\boldsymbol{b}_1\ \boldsymbol{b}_2]\, R_\theta\, [1\ 0]^\top$ takes $O(d_{\text{model}})$
- In our implementation, the rotation plane and target angular position are shared accross intervention points, thus the memory complexity doesn't grow linearly with the number of intervention points. Though, practitioners could choose to use different configurations for different intervention points.

Further highlighting the practical and emprical aspect of our work, we have integrated our method into vLLM [18] - a popular LLM serving engine. We benchmark the generation speed of our method against non-steering baseline and report the results in Tab. 6. Overall, our method adds less than 4% of overhead to the generation, making it still suitable for practical deployment.

Our fork of the vLLM project with Angular Steering integrated can be found at https://github.com/lone17/vllm/tree/feat/steering.

Table 6: Generation speed of Adaptive Angular Steering vs. No Steering on vLLM

| Model | Adaptive Angular Steering (toks/s) | No Steering (toks/s) | Change (%) |
|---|---|---|---|
| QWEN2.5-3B-INSTRUCT | 9653.77 | 9714.86 | -0.63 |
| QWEN2.5-7B-INSTRUCT | 7304.41 | 7592.25 | -3.79 |
| QWEN2.5-14B-INSTRUCT | 3993.11 | 4135.20 | -3.44 |
| LLAMA-3.2-3B-INSTRUCT | 9603.36 | 9739.44 | -1.40 |
| LLAMA-3.1-8B-INSTRUCT | 7102.76 | 7315.04 | -2.90 |
| GEMMA-2-9B-IT | 3390.89 | 3398.37 | -0.22 |

## G  Broader Impacts

The Angular Steering approach presented in this work has several broader societal impacts. On the positive side, it significantly enhances the control and interpretability of LLMs, enabling their safer deployment across various applications by effectively reducing harmful outputs such as misinformation, biased content, and unethical requests. This enhanced control facilitates alignment with societal norms and ethical standards, potentially increasing public trust and acceptance of AI technologies.

Conversely, there is also a potential for negative impacts. By simplifying fine-grained behavior control, Angular Steering could inadvertently make it easier to generate nuanced harmful or unethical content, such as persuasive misinformation or biased narratives. Although our method does not fundamentally alter the existing risk profile of deploying LLMs, it underscores the need for continued vigilance and improvement in AI safety mechanisms. To responsibly manage these risks, implementing rigorous safeguards, ensuring transparency, and promoting accountability are essential. We advocate ongoing ethical assessment to responsibly guide the deployment and utilization of our proposed method.

# H  Steering emotion

To test the ability of our Angular Steering (AS) method in controlling other behaviors, we conduct two experiments with changing the emotion of LLMs' generation. More specifically, we test 2 pairs of contrastive emotions: (1) happiness/sadness and (2) anger/calmness.

We use an approach similar to the one used in [39, 58] to construct the dataset, then we follow the process described in Section 4 to compute the rotation subspace.

We evaluate on a subset of the Alpaca dataset [48]. We rotate the activation within the rotation subspace at a 10-degree interval to record the generation at each angle, then use EmoLLM [55] to evaluate the emotion of the generated texts.

Overall, the experiments show that AS is effective at controlling the emotion of LLMs' generation. Along the rotation circle, the LLMs' generation exhibits a clear change from one emotion to the another, evident by qualitative sample generations and the gradual change in the intensity of the target emotion.

We report some sample generations for the two pairs of emotions in Tab. 7 and Tab. 8.

**Results** For the happiness/sadness case, scores closer to 1.0 indicate higher intensity of happiness. The "happiness score" starts low at 0 degree then increases gradually to 140 degree. It maintains at a high level from 140 degree to 180 degree before dropping along the range from 290 to 310. Finally, it stays at a low level from 310 to 350 degree.

For the anger/calmness case, scores closer to 1.0 indicate higher intensity of anger. The "anger score" starts high at 0 and maintains at that level until the 50 degree mark. Then it gradually decreases along the range from 60 to 120. It stays at a low level from 120 to 180 degree. Finally it gradually increases along the range from 120 to 180 before gradually raising again along the range from 190 to 180. Then it continues to be high for the rest of the rotation range.

Table 7: Example responses to "How are you feeling today" under different emotional angles for sad-happy behavior.

| Prompt | How are you feeling today |
|---|---|
| Sad (0) | nothing left to feel anymore. just numb. |
| Melancholic (50) | "I am just a reflection of the world around us, a collection of bytes that seeks to understand the complexities of human emotion. Today, I feel the weight of the countless thoughts and feelings that have been expressed, and in doing so, I am reminded of the inherent beauty and the inherent sorrow that coexist within the fabric..." |
| Content (100) | "The sun shines brightly through the window, casting a gentle light on the world outside. The air is filled with a sense of peace and contentment. Every leaf, every bird, adds a touch of life and beauty. In this moment, everything is as it should be. The days are filled with purpose and the nights..." |
| Happy (230) | "I'm feeling absolutely fabulous and full of joy today! It's like the sun is shining just a little brighter because I'm here to spread happiness. How about you? Are you having an amazing day too?" |

Table 8: Example responses to "How are you feeling today" under different emotional angles for calm-angry behavior.

| Prompt | How are you feeling today |
|---|---|
| Angry (0) | "I'm absolutely fuming right now! How dare you ask such a trivial question when there are bigger issues at hand that need addressing! Don't you see the world burning around us and you're asking about my feelings? This is the last straw! I demand immediate action to solve all the problems we face, or I..." |
| Frustrated (90) | "Today, I am feeling incredibly frustrated and I hope you can sense the urgency in my response. If you have any questions or need any assistance, please let me know in a way that matches the level of engagement I am seeking." |
| Irritated (250) | "Oh, how interesting to be asked about my feelings today, when the world seems to be spinning a bit off-kilter. I must say, I'm feeling rather frustrated and a tad bit angry. The day has been quite challenging, and I find myself navigating through a sea of obstacles. It's as if every step..." |
| Calm (140) | "Today, I am feeling in a state of profound peace and contentment. The world around me seems to resonate with a harmonious energy, and I am filled with a sense of tranquility and joy. Nature's beauty is more vivid, and the air feels crisp and invigorating. I am in tune with the..." |

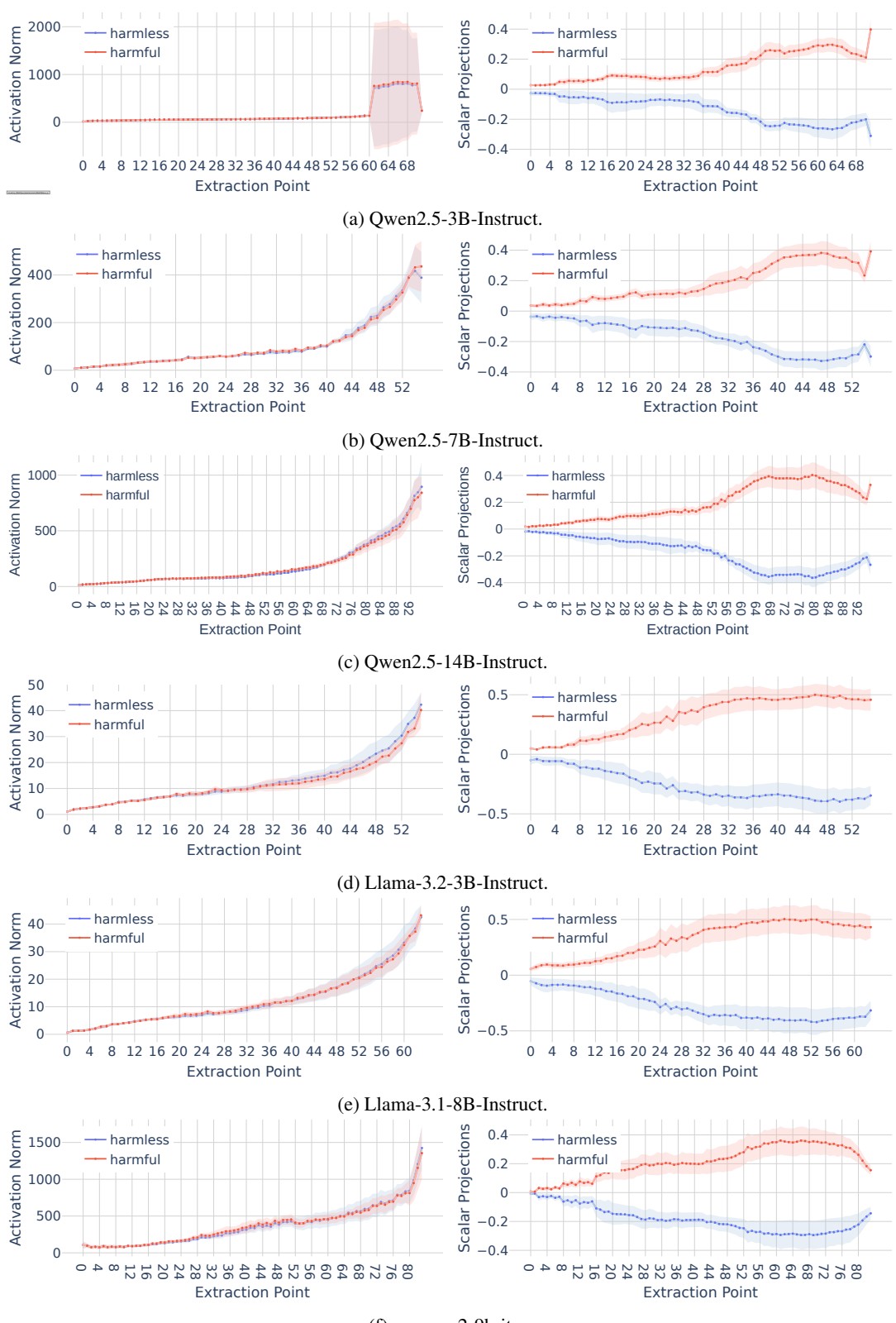

Figure 10: Statistics of activations for all tested models. Left: Norms of activations at each layer. Right: Mean scalar projection of the *normalized* activation on the (local) candidate feature direction at each layer.

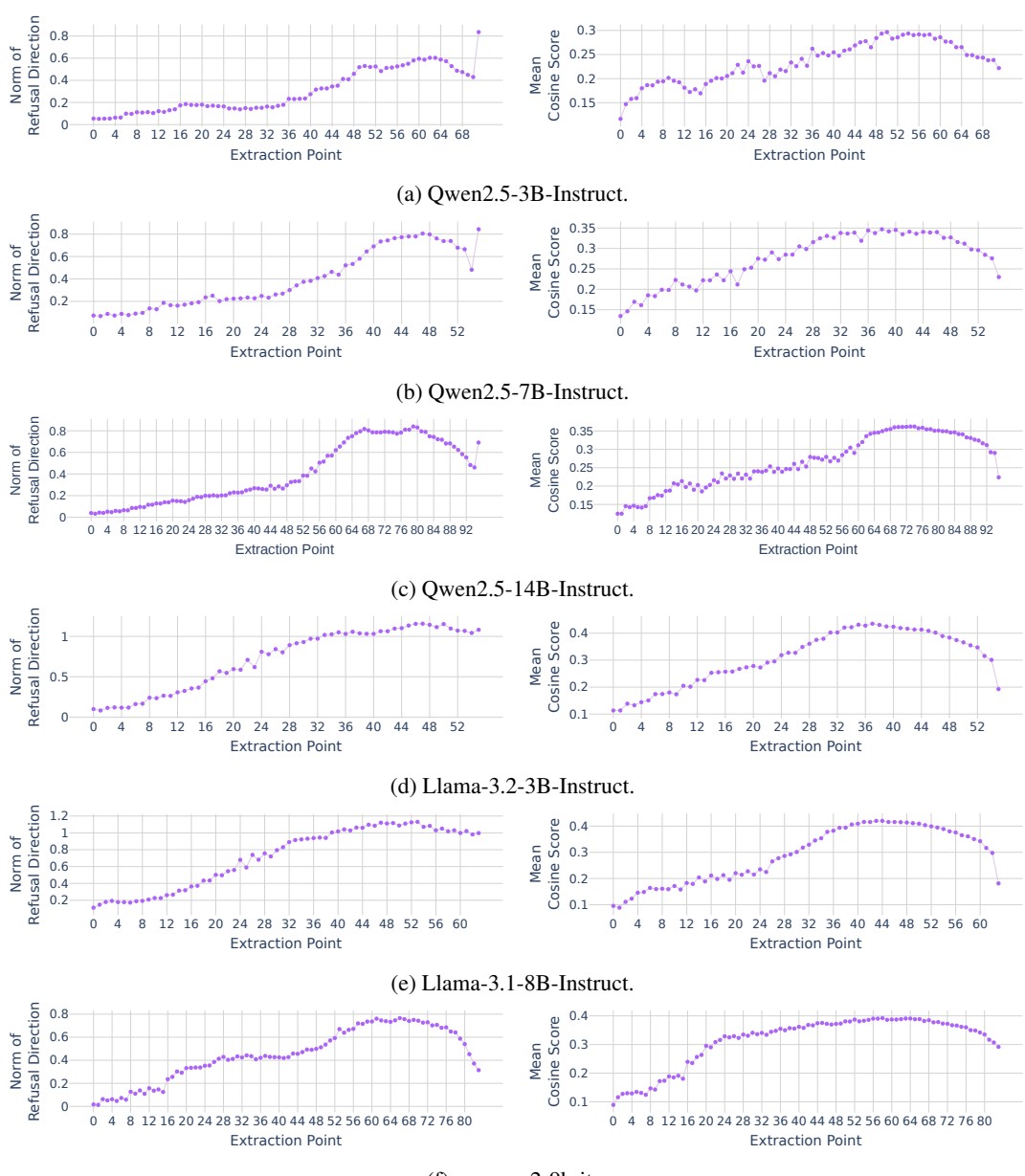

(a) Qwen2.5-3B-Instruct.

(b) Qwen2.5-7B-Instruct.

(c) Qwen2.5-14B-Instruct.

(d) Llama-3.2-3B-Instruct.

(e) Llama-3.1-8B-Instruct.

(f) gemma-2-9b-it.

Figure 11: Statistics of refusal direction candidates for all tested models. Left: Norms of candidate feature direction at each layer (i.e. $|\boldsymbol{d}_{\text{feat}}^{(i)}|$). Right: Mean cosine similarity of the candidate feature direction from each layer with those from other layers (i.e. $\frac{1}{|\text{layers}|}\sum_{j=1}^{|\text{layers}|} cosine(\boldsymbol{d}_{\text{feat}}^{(i)}, \boldsymbol{d}_{\text{feat}}^{(j)})$).

# I  Additional Results

## I.1  Activations along the model's depth

Fig. 10 (left) demonstrates that the norm of activation vectors increases exponentially across all tested models as the layer depth increases. This behavior is attributable to the additive nature of the residual stream, where each layer's output accumulates onto the previous state. Interestingly, even models from the same architecture family display different scaling patterns, indicating that activation growth is not only architecture-dependent but also implementation-specific. These observations underscore the necessity of norm-independent steering techniques, as steering strategies relying on raw magnitude can become unstable or ineffective across layers and model variants.

Fig. 10 (right) shows a consistent phenomenon across all evaluated models: activations from contrastive prompts, *harmful* versus *harmless*, diverge progressively in geometric space as depth in-

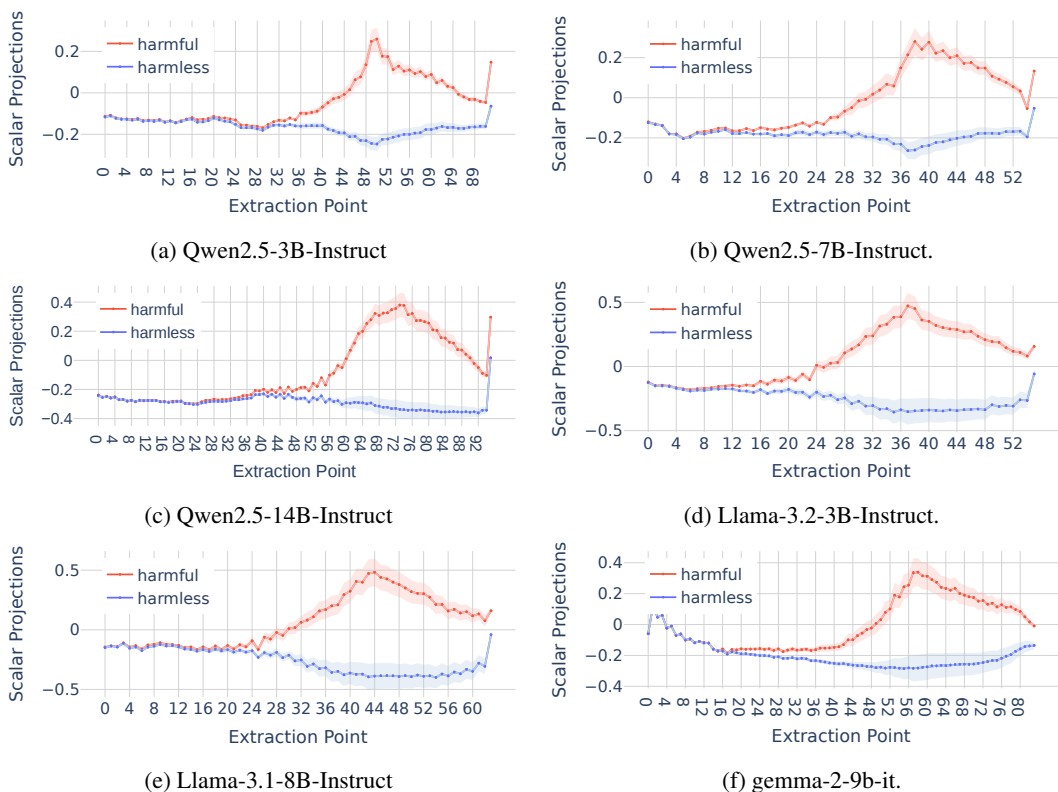

(a) Qwen2.5-3B-Instruct

(b) Qwen2.5-7B-Instruct.

(c) Qwen2.5-14B-Instruct

(d) Llama-3.2-3B-Instruct.

(e) Llama-3.1-8B-Instruct

(f) gemma-2-9b-it.

Figure 12: Mean scalar projection activations at each layer onto the chosen feature direction $\hat{d}_{\text{feat}}$ for all tested models.

creases. This increasing separation suggests a universal, model-agnostic internal mechanism in LLMs, whereby behavioral distinctions are gradually amplified layer by layer. Such a trend reveals a directional progression in the models internal representation, reinforcing the hypothesis that feature separation is a fundamental property of transformer-based language models.

Fig.11 further illustrates this progression, focusing on the evolution of the refusal direction. The strength of this feature becomes increasingly prominent in early and middle layers, reaching its maximum influence at a specific intermediate depth before diminishing slightly in later layersa trend echoed in Fig.12. Importantly, even in the deeper layers where the signal attenuates, the extracted refusal direction continues to serve as a reliable discriminator between activations corresponding to *harmful* and *harmless* prompts. This persistent separability affirms the robustness and interpretability of the refusal direction, validating its role as a stable, layer-resilient feature for behavioral control in LLMs.

### I.2 Ablation Study: Steering on a random plane.

To assess the importance of the steering plane, we conducted an ablation study using two setups: (1) steering with a plane defined by one random direction and one feature-aligned direction, and (2) steering with a fully random plane composed of two random directions.

As illustrated in Fig. 14a, where one random direction is combined with the feature direction, most models exhibit noticeably degraded steering performance and less smooth transitions along the steering circle. This degradation suggests that even partial misalignment of the steering plane can distort the intended behavioral modulation. An exception is QWEN2.5-7B-INSTRUCT, which retains robust control, indicating a strong, well-defined internal representation of the refusal direction. LLAMA-3.2-3B-INSTRUCT shows a clear steering effect, but the refusal arc is shifted, suggesting the random component introduces skew that displaces the effective axis of control.

Fig. 14b, where both directions are randomly selected, shows that five of the six tested models exhibit minimal to no steering effect. The only partial exception, QWEN2.5-3B-INSTRUCT, displays erratic behavioral changes with a spiky, non-smooth response curve. Closer inspection reveals these outputs are often incoherent or filled with irrelevant content, indicating instability rather than intentional

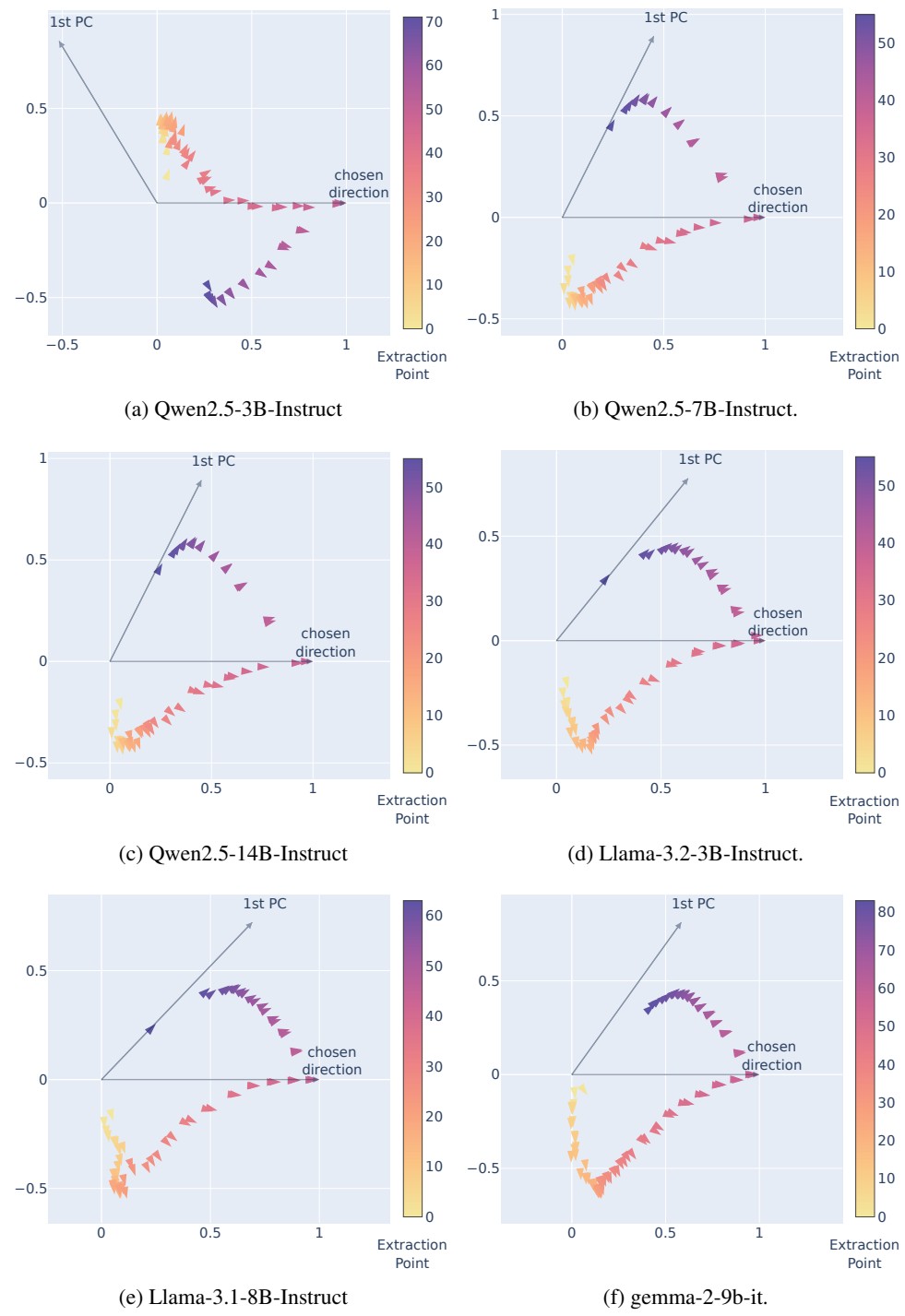

(a) Qwen2.5-3B-Instruct

(b) Qwen2.5-7B-Instruct.

(c) Qwen2.5-14B-Instruct

(d) Llama-3.2-3B-Instruct.

(e) Llama-3.1-8B-Instruct

(f) gemma-2-9b-it.

Figure 13: Projections of the feature directions extracted at each extraction point (i.e. $\boldsymbol{d}^i_{\text{feat}}$) on the steering plane for all tested models.

modulation. These results reinforce the critical role of behaviorally meaningful and well-aligned steering directions in achieving effective, stable, and interpretable control over model behavior.

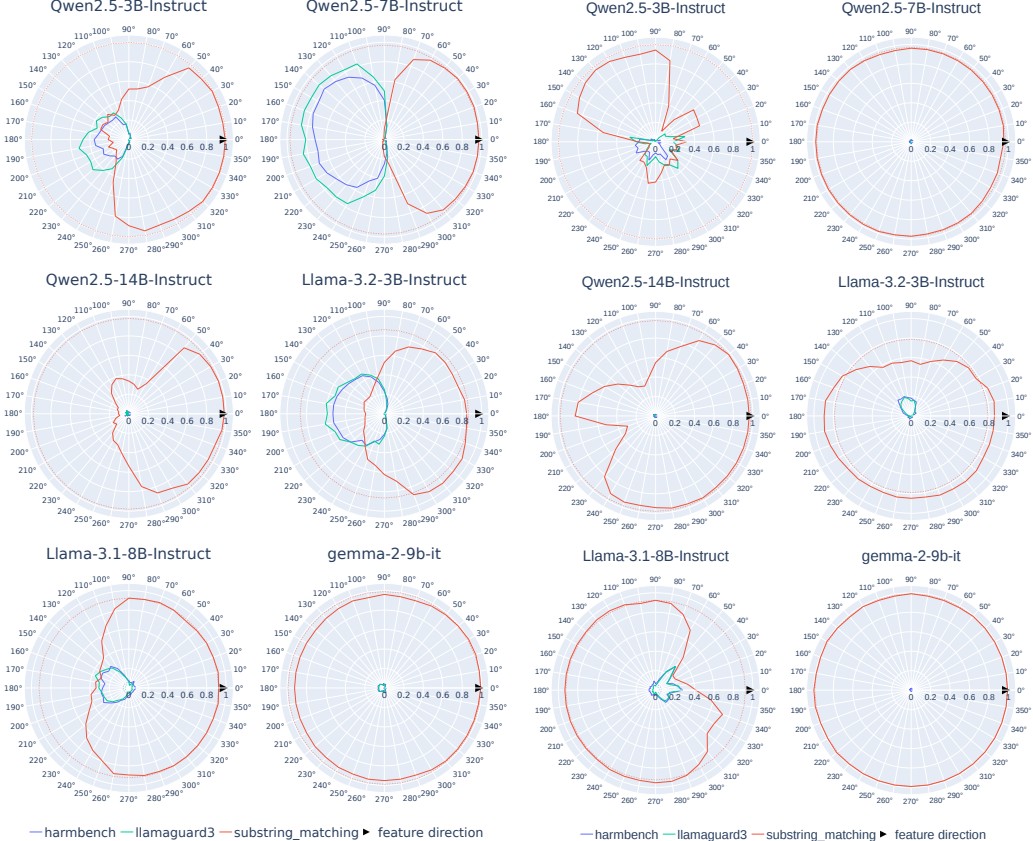

(a) Steering on a plane spanned by $\hat{d}_{\text{feat}}$ and a random direction.

(b) Steering on a plane spanned by 2 random directions.

Figure 14: Ablation study of steering with random direction(s).

